health and disease and epidemiology

disease risk perception,
behavioural epidemiology, social norms,
behaviourally receptive phase

**Author for correspondence:**
N. H. Fefferman
e-mail: nina.h.fefferman@gmail.com

# Improving pandemic mitigation policies across communities through coupled dynamics of risk perception and infection

M. J. Silk[1], S. Carrignon[2,3,4], R. A. Bentley[3] and N. H. Fefferman[5,6]

[1]Centre for Ecology and Conservation, University of Exeter, Penryn Campus, UK
[2]Center for the Dynamics of Social Complexity, [3]Department of Anthropology, [4]School of Information Sciences, [5]Department of Ecology and Evolutionary Biology, and [6]Department of Mathematics, University of Tennessee, Knoxville, TN, USA

 SC, 0000-0002-4416-1389; NHF, 0000-0003-0233-1404

Capturing the coupled dynamics between individual behavioural decisions that affect disease transmission and the epidemiology of outbreaks is critical to pandemic mitigation strategy. We develop a multiplex network approach to model how adherence to health-protective behaviours that impact COVID-19 spread are shaped by perceived risks and resulting community norms. We focus on three synergistic dynamics governing individual behavioural choices: (i) social construction of concern, (ii) awareness of disease incidence, and (iii) reassurance by lack of disease. We show why policies enacted early or broadly can cause communities to become reassured and therefore unwilling to maintain or adopt actions. Public health policies for which success relies on collective action should therefore exploit the *behaviourally receptive phase*; the period between the generation of sufficient concern to foster adoption of novel actions and the relaxation of adherence driven by reassurance fostered by avoidance of negative outcomes over time.

## 1. Introduction

The ongoing COVID-19 pandemic has cemented the understanding among public health researchers, practitioners and policy makers that the spread of infectious disease is more than a purely epidemiological process. While COVID-19 has strained hospital capacity [1], the global supply of personal protective equipment [2,3], food supply chains [4,5] and unemployment insurance processing capacity [6], the greatest challenges in understanding, predicting and planning mitigation for the ongoing spread of the disease lies in how to understand, anticipate and influence human behavioural responses [7].

Efforts to incorporate social [8], psychological [9] or economic [10] factors have revealed the profound effects of behavioural choices on projected outbreak dynamics [11,12]. Critically, however, many studies helping shape policy have considered behavioural factors as mostly uniform across the affected population and mostly constant throughout the course of an outbreak [13]. While there are legitimate and important reasons to explore models making these assumptions, they do not reflect the current reality of behavioural responses to the COVID-19 pandemic.

People's behavioural responses to the pandemic will vary considerably, often between locations and over time, driven by variations in local government policy as well as individual behaviours over time (e.g. as 'stay at home' orders were enacted and relaxed; [14]), and across social and demographic groups (e.g. conservative versus liberal; [12,15]). These are not static parameters; these shifting patterns of behaviour are not independent of the spread of infection but are instead inextricably coupled with the spatial and temporal patterns of disease incidence.

Models that have incorporated individual behavioural responses with the epidemiological dynamics of an ongoing outbreak (e.g. avoiding sick people, accepting a vaccine; [16–22]) provide the basis for developing more dynamic approaches that allow individual behaviours to change when some psychological threshold for change is met. Understanding, analysing and predicting such coupled dynamics, in which behavioural responses themselves shift over space and time, will allow us to build outbreak models that can apply broadly across regions and withstand shifting conditions for more than a few weeks at a time. Such models can then support policy makers in designing flexible, responsive plans that can be better communicated to, and accepted by, the public, improving efforts to mitigate current risks and helping us prepare for future pandemics.

Amenable to this dynamic behavioural approach, network models in epidemiology have been used with great success to understand how individual heterogeneity in contacts among individuals can cause deviations in outbreak progression relative to homogeneous, mean-field approximations of average behaviours [23]. Analogously, network models of 'social contagions' predict the spread of beliefs or information through populations [24–29]. Studies focusing on specific aspects of social contact networks have shown that a variety of structural features—edge density, clustering coefficient and modularity—of those networks affect the progression of epidemics and information [30–36].

For social-behavioural phenomena, multiplex (also called multilayer) networks capture ongoing, coupled dynamics (e.g. [37–39]). In application to coupled behaviour-epidemiological dynamics, multiplex networks can be used to combine a physical contact, or 'infection', network over which an infectious disease might be transmitted and a communication network over which information or opinions might be shared. They can therefore provide the required tool for coupling the states and dynamics between the layers [40,41]. Consequently, simulations exploiting coupled multiplex networks have provided important insights into the impact of social construction of risk perception on the spread of infectious diseases (e.g. [42,43]).

Social distancing exemplifies how coupling between a communication and infection network layer is critical to the COVID-19 pandemic. Adherence to social distancing recommendations is determined by individual concern and resulting behaviours. Those concerns are, in turn, constructed by each individual based both on direct observation of evidence (e.g. contact or lack of contact with sick people), and on social inputs, including communication with worried peers/advisors and perception of social norms (e.g. knowledge of the compliance of others). These social inputs rely on a communication network which may or may not overlap with the contact network over which infection spreads. The communication network and contact network can be considered as layers of a multiplex network containing the same set of interacting individuals. Coupled dynamics between the communication and infection layers of the multiplex network can have some profound impacts on the progression of epidemics and the efficacy of attempted mitigation strategies. Characteristics of the communication layer will impact which individuals and communities perceive the risk to be high (whether or not the actual risk of infection is high in their physical environment), which will influence rates of adherence to disease-defensive public health policies. For the same reasons, the multiplex structure should influence when disease permeates heterogeneous networks, reaching different communities at different time/stages of the outbreak, and how rapidly it spreads through each community when it does reach them, especially as increased disease prevalence should increase perceived risk, and thereby slow the progression via behavioural defensive responses. Two of the most likely network features to be important in shaping the interplay between these dynamics are homophily (the tendency to be affiliated with similar others) [44–47] and modularity (the strength of division between 'communities'; populations more likely to be connected to each other than to individuals outside of the group) [48,49].

To begin to understand how these coupled behavioural-epidemiological dynamics may be driving the current COVID-19 pandemic, we here present a multiplex network model that captures a standard susceptible–exposed–infected–removed (SEIR) epidemiological dynamic [50], in a population with a simplified age structure (children, adult and elderly), and with two opposing 'predispositions' that contribute to homophily in either the infection layer of the network, communication layer or both. Our model assumes that social influence can be effective at elevating individual concern about disease, but only direct observation of an absence of illness in a community will provide reassurance (i.e. cause a decrease of concern). We assess the severity of disease progression by a joint measure considering both the time and magnitude of the epidemic peak in each community. We explore the relative impact of social versus observational estimation of disease risk on the epidemic outcomes in communities, the role of reassurance on the likely dynamics in risk behaviours over time and the influence of modularity and homophily according to predisposition on these patterns. While this model is parameterized to reflect current COVID-19 features and challenges, it seems clear that socio-behavioural dynamics that shape the nature of risk perception, and therefore disease protective behaviours should be important for any pandemic preparedness planning for the future. We therefore present a broad set of scenarios.

## 2. Methods

### (a) Overview

We used stochastic simulations to test how the spread of concern influenced epidemic dynamics of COVID-19 through populations divided into communities in both communication and physical contact structures (figure 1). All simulated populations consisted of 2000 individuals of three age-categories: children (18 or younger), young (low-risk) adults (19–64) and old (high-risk) adults (65 or older) and split between two distinct 'predispositions'. Predispositions could be used to reflect any characteristic or belief that may influence the tendency of individuals to be connected in either the communication layer, infection layer or both (a relevant example being political views in the USA). We then designed realistic, social networks for our populations and used these coupled (multiplex) networks to determine how infection and the spread of concern about the disease can spread through the population and interact (figure 2). We recorded epidemic outcomes for a range of network structures and proposed models for the change in concern of individuals over time. All modelling was conducted in R 3.6.1 [51]. R code used is provided in the electronic supplementary material and on GitHub (https://github.com/matthewsilk/CoupledDynamicsNetworkPaper/). Full methods are detailed in the electronic supplementary material.

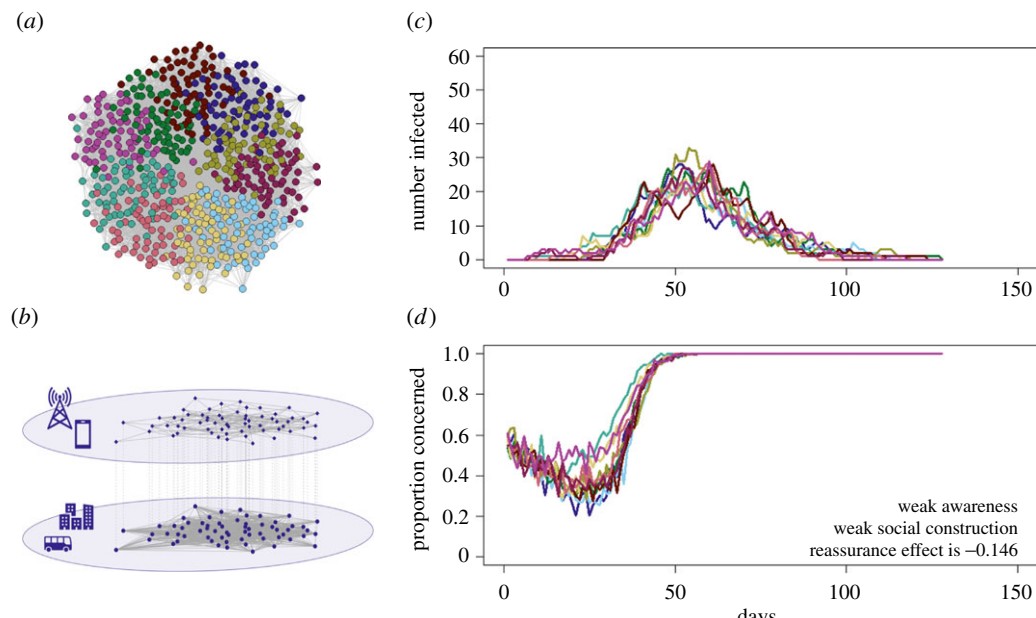

**Figure 1.** A depiction of the modelling process. We generated coupled multiplex networks (*a,b*) containing a communication and infection layer for 2000 individuals. This network was divided into 10 equally sized social communities and the network was rewired to have a desired modularity (*a*). Here (*a*) we show community assignment in the infection layer for young adults of a single predisposition (i.e. only part of the network). (*b*) The infection layer and communication layer of the multiplex network differed in density so that the infection layer was better connected and could have the same or higher modularity and the same or lower homophily according to predisposition. We then modelled the spread of infection (*c*) and change in adherence to social distancing (*d*) and recorded these at a community-level. The colours of the lines depict the community membership illustrated in (*a*). (Online version in colour.)

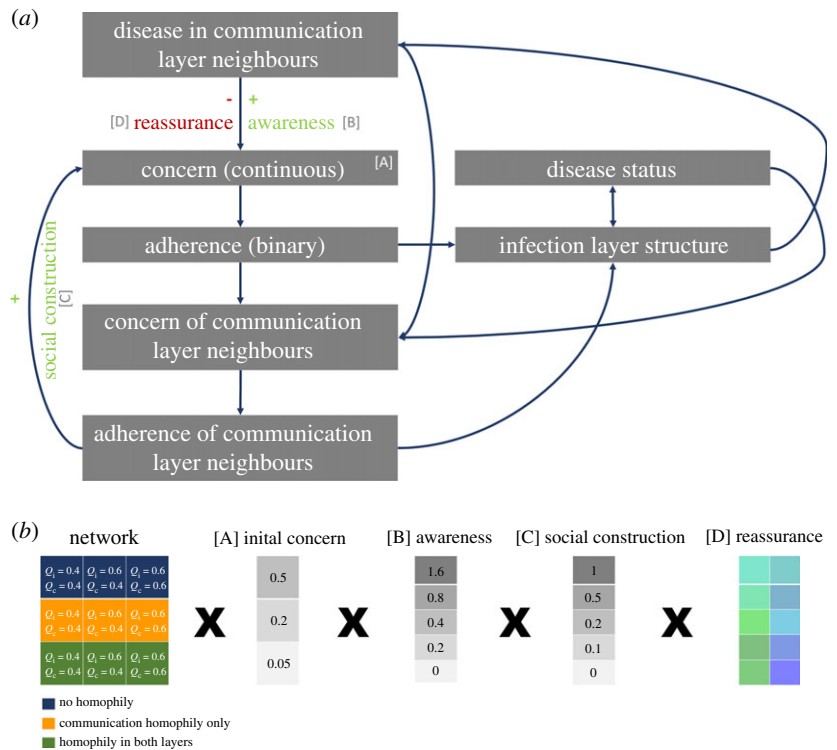

**Figure 2.** (*a*) A schematic providing an overview of how our stochastic model couples risk perception and infection dynamics and (*b*) details on the key parameters varied during the simulations. Values of the reassurance effect were drawn from a uniform distribution between −0.2 and −0.01. In the figure, we refer to the modularity of the communication layer at $Q_c$ and the modularity of the infection layer as $Q_i$. (Online version in colour.)

## (b) Population generation

Our population was 24% children, 63% young adult and 13% old adult to match recent United States demographic data. Age classes could differ in the social connections, epidemiological outcomes and concern about the disease (as detailed in relevant sections). Individuals also had one of two baseline predispositions: 'A' and 'B'. Fifty per cent of individuals were

of each predisposition. Homophily according to predisposition could impact social connections.

## (c) Social network generation

We generated nine multiplex social networks that connected all 2000 individuals within a 'communication' layer that influenced the spread of concern about the disease and an 'infection' layer

that influenced the transmission of the pathogen itself. Each network layer consisted of 10 equally sized social communities (community size: 200). We simulated equally sized communities as a simplifying assumption to focus on the importance of the learning processes of interest. Networks were simulated in a stepwise fashion so that connections in both the communication layer and infection layer displayed community structure, homophily by age and could display homophily by predisposition (see the electronic supplementary material, Methods). Children were connected to parents in the young adult layer and shared their parents' connections with older adults. We used three different combinations of modularity and three different combinations of homophily by predisposition. Using relative modularity [48], sections of our networks either had a modularity (based on pre-defined social communities) of 0.4 in both layers, a modularity of 0.6 in both layers, or the infection layer had a modularity of 0.6 and the communication layer a modularity of 0.4 (to reflect the fact that more communication is likely between communities than contacts relevant for infection). The proportion of each predisposition (A and B) within each community was the same as that in the overall population. We made this decision as a simplifying assumption (to focus our analyses on the learning processes of interest), although in reality social communities could be biased towards particular predispositions. For homophily by predisposition, we included networks in which there was either (i) no homophily in either layer, (ii) homophily in the communication layer only, or (iii) homophily in both layers.

## (d) Concern model

We modelled the spread of concern about the disease through the communication layer as a complex contagion [52]. For all adults, whether an individual was adherent or not (binary trait) depended on a Bernoulli draw based on an underlying probability which we term concern (for information on the adherence of children see the eectronic supplementary material, Methods). Consequently, it was possible for individuals to move from being non-adherent to adherent but also for them to return to being non-adherent. Individuals with intermediate levels of concern were likely to fluctuate between adherent and non-adherent states. Initial levels of concern correspond to a 50, 20 or 5 chance of adherence. The underlying concern of all adults could then be influenced by social construction, awareness or reassurance (figure 2a). We focused on the region of parameter space where social construction could only increase concern to isolate the importance of the reassurance effect (i.e. to be conservative relative to our effect of interest). However, in some contexts, social construction could lead to reductions in concern and adherence in its own right.

Concern and adherence were modelled at an individual level throughout each simulation. Each time an individual became adherent, individuals cut their connections within the infection layer of the network while maintaining their connectivity in the communication layer. Individuals cut all non-parent-child connections with a 50% probability (selected to represent a reasonable general approximation of a real-life effect across diverse demographic and socio-economic groups; see the electronic supplementary material, Methods). Connections were cut to have an edge weight of 0.001 meaning that the probability of transmission across them was negligible. Old adults that cut connections with young adults also cut connections with their children. If an individual became non-adherent then these edge weights returned to their initial values.

## (e) Infectious disease model

We modelled the spread of SARS-CoV-2 using an age-structured stochastic SEIRD model adapted from [50] and adjusted to match empirical data as detailed in the electronic supplementary material, Methods. The model contains susceptible (S), exposed (E), mildly or pre-symptomatic (I1), symptomatic (I2), hospitalized (I3), recovered (R) and dead (D) compartments. The transition between compartments is detailed below and parameter values are provided in the electronic supplementary material, table S6. Symptomatic (I2) and hospitalized (I3) individuals cut all connections (including to children) in the infection layer of the network to 0.001, meaning that individuals are only likely to spread infection during the time they are I1 (mean: 4 days). These connections were restored to their full value if and when individuals recovered.

## (f) Simulations

For each of the nine multiplex networks studies we conducted simulations for the full combination of starting values (3), social construction effects (5) and awareness effects (5). For each of these 75 combinations, we conducted 10 replicate simulations with the same network structure but with different individuals seeded with infection (figure 2b). A unique value of the reassurance effect was drawn from the uniform distribution defined above for each run of the simulation (figure 2b). We simulated a time period of 300 days (or until there were no remaining infected individuals). The simulation algorithm is detailed in the electronic supplementary material, Methods.

## (g) Analysis

To test the combined effects of the social construction of concern, increases in concern owing to awareness of incidence of disease, and relaxation of adherence to social distancing owing to reassurance effects from not knowing infected people, we first calculated a measure of epidemic severity that took into account the height and timing of the epidemic peak in each community (figure 3):

$$\text{severity} = \frac{\text{height of epidemic peak}}{\text{max height of epidemic peak in all simulations}} \times \left(1 - \frac{\text{time from start of outbreak to epidemic peak}}{\text{max time from start of outbreak to epidemic peak}}\right),$$

(see also the electronic supplementary material, Methods)

We then used linear mixed effects models to ascribe variation in epidemic severity to network structure and values of the social construction, awareness and reassurance effects as detailed in the electronic supplementary material, Methods.

## 3. Results

We monitored the epidemiological outcomes and changes in adherence to protective behaviours (e.g. social distancing) over time separately for each of 10 communities or modules in our social networks (figure 1).

## (a) Communities infected later are often hit harder

In the absence of any changes to concern or adherence (which we term learning for convenience), communities hit later in the epidemic typically have more severe outbreaks than those hit earlier (figure 4; electronic supplementary material, figure S2 and tables S1–S5), with a reduced time from the start of an outbreak to its peak, and a higher peak (electronic supplementary material, figure S1). This pattern remains regardless of whether initial social distancing levels are high (50% of the population adherent; electronic supplementary material, table S1) or intermediate (20% adherent; electronic supplementary material, table S2). These patterns were qualitatively similar across all of the nine multiplex networks simulated. However, the overall models estimated that outbreaks were typically less severe when there was matching homophily in the infection and

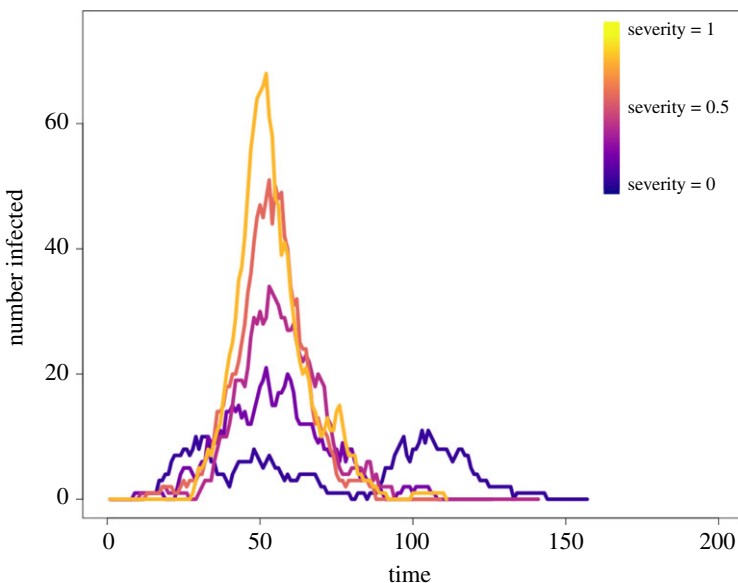

**Figure 3.** An illustration on the relationship between our measure of epidemic severity (line colour) and the shape of the epidemic curve in a community for five selected simulation runs. (Online version in colour.)

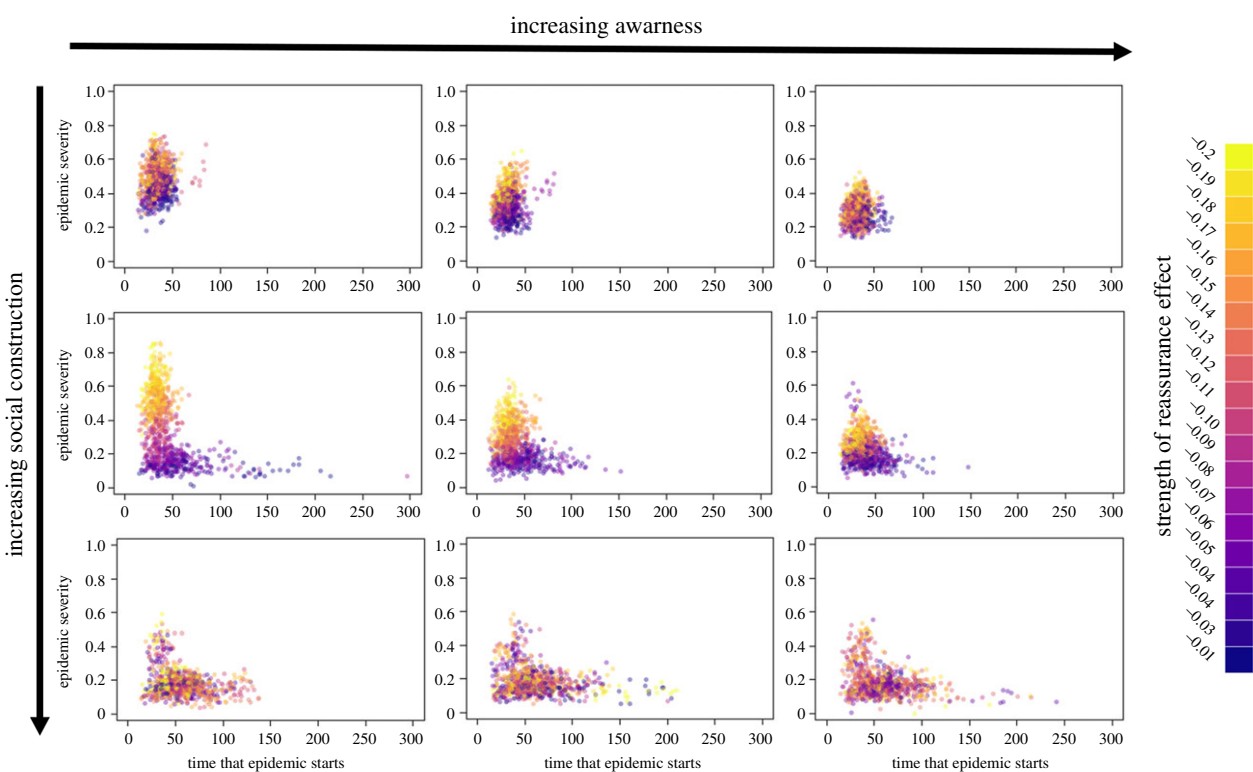

**Figure 4.** The relationship between the time that an epidemic starts in a community and the severity of that epidemic for a range of different awareness effects (0, 0.4, 1.6) and social construction effects (0, 0.2, 1). In the top row, there is no change in the concern of individuals through social construction, while in the left-hand column, there is no effect of awareness. Conversely, these two parameters take their maximum values in the bottom row and right-hand column, respectively. Points are coloured on a continuous scale according to the strength of the reassurance effect. For each run of the simulation, this was sampled from a uniform distribution between −0.2 and −0.01. (Online version in colour.)

communication layers, and more severe when there was homophily only in the communication layer (electronic supplementary material, tables S1–S2).

## (b) Different forms of learning flatten the curve in different ways

Increased adherence to social distancing through awareness of ill individuals and through social construction of concern both reduced epidemic severity but also resulted in strikingly different epidemic outcomes (figure 4; electronic supplementary material, figure S2). When individuals increased adherence owing to awareness, the height of the epidemic peak was reduced but the time to reach the peak was not (electronic supplementary material, figure S1). By contrast when individuals became more likely to be adherent to social distancing through social construction both the height of the epidemic peak and the length of time between the start of an outbreak and epidemic peak increased. This resulted in the timing of epidemic peaks becoming more variable between communities (electronic supplementary material, figure S3) which may have important implications for mitigation.

(a)

(b)

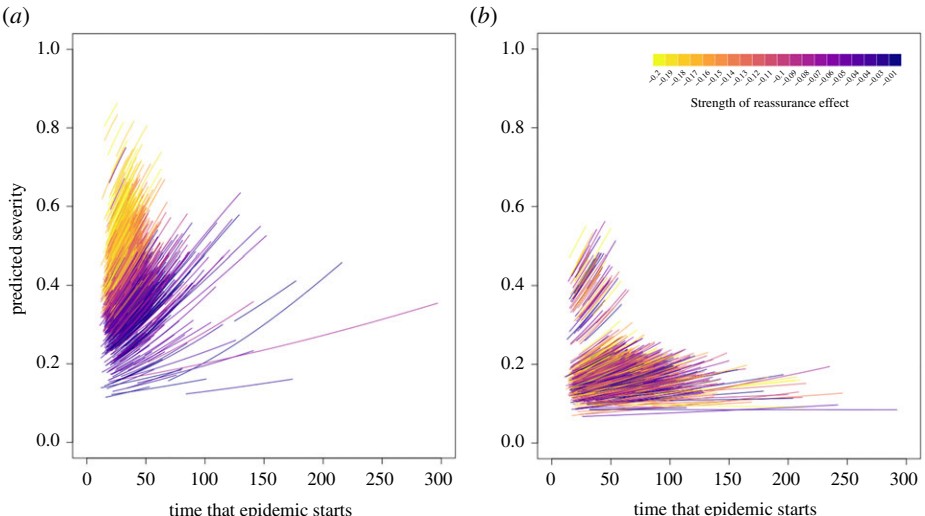

**Figure 5.** Random effect estimates for the effect of each simulation run from the statistical model fitted to explain epidemic severity in the absence of a reassurance effect. Lines are plotted using estimates for the random intercept and random slope for when (a) social construction is weak (0, 0.1, 0.2) and (b) social construction is strong (0.5, 1). Lines are coloured on a continuous scale according to the strength of the reassurance effect. For each run of the simulation, this was sampled from a uniform distribution between −0.2 and −0.01. Yellow indicates a strong reassurance effect and purple indicates a weak reassurance effect. For the purposes of plotting the lines, we assume the social construction effect is fixed at 0.1 (low) for (a) and fixed at 1 (high) for (b). We assume no awareness effect and use estimates that apply to network ID 1 (modularity of 0.4 for both layers and no homophily according to predisposition). (Online version in colour.)

Social construction (responding to the adherence of communication neighbours) had a much greater impact on mitigating epidemic severity than the awareness effect (responding to the illness of communication neighbours). However, low levels of social construction were much less effective (figure 4) with only moderate influence of adherent neighbours having a substantial mitigating effect when initial levels of concern were lower. By contrast, increases in the impact of awareness had a more linear effect on epidemic severity (figure 4). Results were qualitatively similar regardless of the modularity or homophily of the network.

## (c) Social construction of concern is particularly important for later-hit communities

Increased adherence through social construction, and to a lesser extent awareness, helped later-hit communities disproportionately. Rather than being more severe, outbreaks in these communities were no worse or even less severe than those in communities hit earlier (electronic supplementary material, tables S1–S2) when social construction was strong. Similarly, to the previous result, this outcome was weaker and/or less likely to be achieved with only weak social construction (electronic supplementary material, tables S1–S2). The importance of social construction in preventing more severe outbreaks in communities hit later was greatest when the infection layer of the multiplex network was more modular ($Q_{rel} = 0.6$), i.e. in the cases where the steepest increase in epidemic severity over time would have occurred in the absence of social construction (electronic supplementary material, tables S3–S5). We also found some evidence that the mitigating effects of social construction and awareness were more likely to overlap (cancel each other out to some extent) when there was matching homophily and high modularity ($Q_{rel} = 0.6$) in both layers or when there was no homophily but the modularity of the communication layer was lower than that of the infection layer (electronic supplementary material, tables S3–S5).

## (d) The reassurance effect when communication neighbours are healthy amplifies differences

In the absence of social construction, the reassurance effect of people becoming more relaxed in their probability to adhere to social distancing over time is critical for epidemic dynamics. When social construction of concern is absent, or only has a weak effect, then a strong reassurance effect causes outbreaks within communities to be more severe on average and disproportionately impacts later-hit communities so that outbreaks tend to be much more severe (figure 5a). It also reduces variability in outbreak peaks between communities (electronic supplementary material, figure S4) resulting in greater population-level synchrony in epidemic dynamics.

However, when social construction is stronger, the link between epidemic severity and the reassurance effect is no longer present (figure 5b). Similar outcomes are observed even when the tendency to adhere would otherwise decline quickly over time in the absence of knowing anyone who is sick.

This result is caused by there being a very strong relationship between epidemic severity and the minimum proportion of each community adherent to social distancing (figure 6; electronic supplementary material, figure S5). When people pay more attention to the concern of their network neighbours, the proportion of non-adherent people remains low even in later-hit communities (figure 6; electronic supplementary material, figure S5). As above, the effectiveness of social construction in preventing the adverse effects of reassurance has a steep threshold, meaning that maintaining the importance of social norms throughout an outbreak is important.

After controlling for other factors, the correlation between the strength of the reassurance effect and both epidemic severity and the increase in epidemic severity over time was relatively consistent between the nine networks modelled (electronic supplementary material, figures S6–S7). The correlation was weakest in network 2 when the infection layer and communication layer were both more modular ($Q_{rel} = 0.6$), and there was no homophily according to

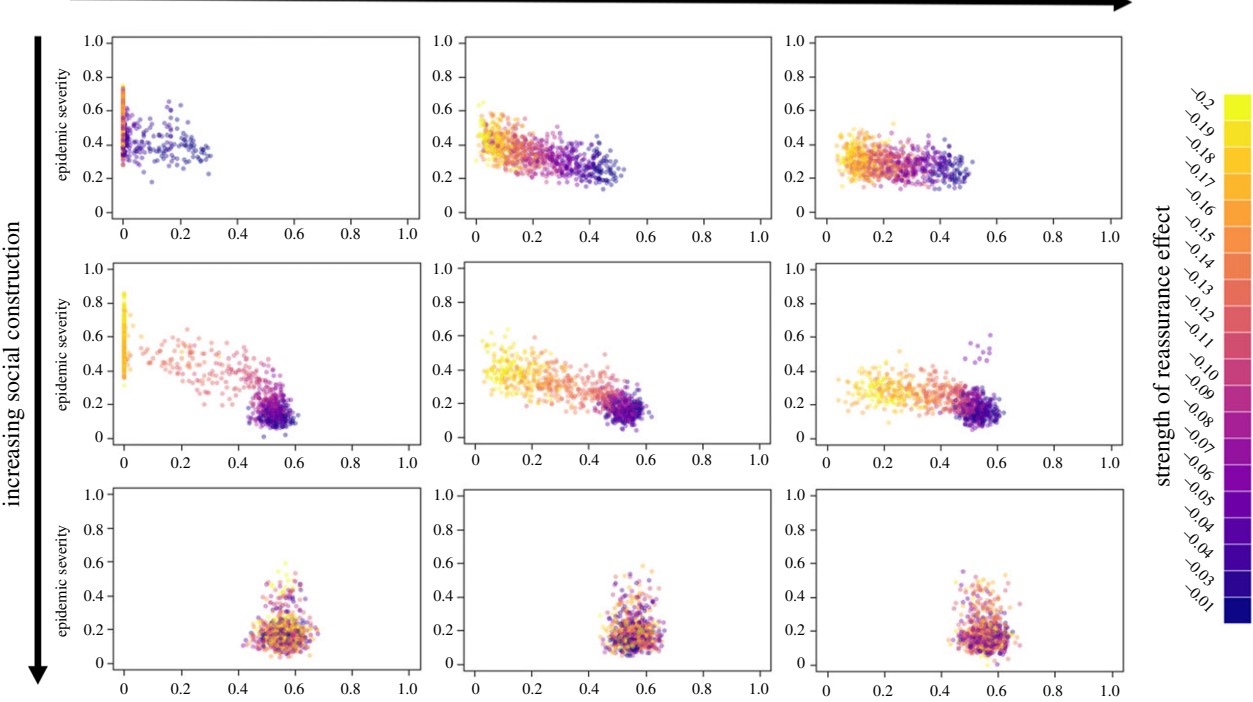

**Figure 6.** The relationship between the minimum proportion of people adherent with social distancing in a community and the severity of that epidemic for a range of different awareness effects (0, 0.4, 1.6) and social construction effects (0, 0.2, 1). In the top row, there is no change in the concern of individuals through social construction, while in the left-hand column there is no effect of awareness. Conversely these two parameters take their maximum values in the bottom row and right-hand column, respectively. Points are coloured on a continuous scale according to the strength of the reassurance effect. For each run of the simulation, this was sampled from a uniform distribution between −0.2 and −0.01. (Online version in colour.)

predisposition (i.e. people interacted with people of the same and different predispositions equally). Both of these features could potentially mean an individual's perceived risk more accurately matches the actual local prevalence of infection.

## 4. Discussion

Our results clearly show the importance of all the considered mechanisms (awareness of infection, social construction and reassurance) in driving concern and shaping the ongoing dynamics of disease progression through distinct social communities. Understanding how each mechanism acts, both independently and synergistically, will be of critical importance in accurately anticipating infection risks in large, heterogeneous populations. One clear, unfortunate impact of effective protective behaviours that serve to buffer or insulate a community from early disease spread is that they can foster social reassurance, degrading concern in the need to maintain them [53]. It may therefore be that, in the absence of further interventions, communities hit later in the outbreak may be paradoxically less well aligned towards behavioural defences than if those communities had experienced greater numbers of infections earlier on.

For the spread of COVID-19, these results indicate the potential for a 'perfect storm'. The delay between behavioural response and local increase in disease prevalence hinders concern-based protective behaviours, whether based on observational awareness or social construction. The infection network structure of the United States is itself a modular patchwork centred around towns and cities of vastly different population sizes [54]. The corresponding communication network is also clearly highly modular and highly homophilic in

its belief structures regarding infection risks and behavioural decisions during COVID-19 [47], and this can extend to political decision-makers themselves [55]. This creates patchy echo boxes of communities that do not experience the same risks at the same times, even while centralized reporting discussed COVID-19 risk nationwide. Consequently, policy such as 'stay at home' orders in populations that had not internalized social norms of increased concern nor experienced sufficiently high levels of infection probably insulated these communities from widespread transmission and allow reassurance to decrease concern, and therefore belief in the need for action. Instead, the normalization of the perceived risk of infection before the actual escalation of local disease incidence slows or prevents the uptake of other protective behaviours when introduced (as has been seen with other disease interventions, e.g. [56]). Initially less affected areas may instead be better served to plan and prepare (both practically and psychologically) to more effectively mount mitigation efforts when disease inevitably arrives.

Our model indicates the value of policies based on the normalization of protective behaviours when the population may be most accepting of, and therefore adherent to them owing to their perceived risk of infection. Timing and scaling the magnitude of interventions to match both individually observable disease incidence and the socially constructed concern for disease risks in each community can reduce the likelihood that interventions are rejected over time and so be more effective in the longer term. Dynamic protective policies are not novel. Models based only on epidemiology and healthcare capacity have already proposed pulsed strategies for 'shelter in place' orders, in some cases suggesting that these pulses could continue even until there is sufficient vaccine coverage to achieve herd-immunity levels of protection

[13]. We propose that such dynamic policies should also monitor and account for community-level concern, as has been found useful in the context of response to natural disasters and threats from climate change [57–59]. Critically, we do not suggest that mitigation efforts should respond to concern-driven demand, but instead that policies can anticipate when they might be well enough received to enable concern-driven acceptance and adherence. Vaccination is an excellent example—MMR inoculation can eradicate regional measles, mumps and rubella risks, but outbreaks in highly developed and resource-rich nations still occur owing to parental refusal of vaccination, with current public health efforts now also focused on increasing vaccine acceptance [60–62]. We propose that policy for non-pharmaceutical interventions should follow the same path and incorporate an explicit focus on the public concern.

One route to promoting more prolonged adherence to recommendations is to lessen the reassurance effect. Once normalized, behaviours may be passively (rather than actively) maintained, meaning that reassurance is less likely to decrease their observance. Strong, reinforced social norms are critical in maintaining a community's adherence to protective behaviours over time. The maintenance of these protective behaviours will remain important in the longer term as the failure to maintain social distancing policies can result in asymmetric epidemic curves and plateaus of high death rates [63]. It is likely that there are sensitive windows during which the perception of risk is high enough to foster the adoption of protective behaviours that can be normalized before reassurance undermines the concern that promotes their adoption. We call this the behaviourally receptive phase. In situations where populations are initially highly concerned then one such phase will occur early in an outbreak (although this may not always be the case). Finding approaches that effectively combat reassurance will be integral to the success of intervening early in an outbreak. Asking community leaders to be vocal about true community risk can support ongoing social construction of concern and strengthen social norms around protective behaviours (i.e. 'maven' effects, as in [27]). Municipal efforts to support families with loved ones who have been diagnosed may meaningfully amplify awareness in ways that combat reassurance while focusing on positive aspects of community rather than individual fear and isolation.

While our model represents an important step in advancing coupled behavioural-epidemiological models, there are limitations to consider. First, we made simplifying assumptions that could be altered to extend the insights provided and test different aspects of coupled behavioural-epidemiological dynamics. For example, we assumed in our study that social construction could only amplify rather than reduce concern. In reality, individuals may reduce their concern if those they communicate with are non-adherent, and this could increase the impact of the reassurance effect and make it harder for social construction to flatten the curve. Similarly, we assumed

different groups in our study (whether different age/risk groups or different predispositions) responded to the local prevalence of infection and adherence in the same way and did not include variability in the ability of individuals to socially distance. Differences in response between these groups could also further amplify some of the effects revealed by our model or cause variation in impact among communities. Real-world communities are also likely to vary, and the impact of differences in their size and/or interconnectedness (we only explored a small number of potential network structures here) is worthy of further investigation. Epidemiologically, our model ignored the role of fully asymptomatic infections (i.e. those who never develop symptoms of disease [64]) in driving disease spread without a concomitant impact in the communication layer. If the probability of remaining asymptomatic is partially age-dependent [65], then communication neighbourhoods among similarly aged individuals may exacerbate differences between community-driven risk perception. Beyond limitations to the model itself, there are obvious, immediate, additional questions that need to be considered. These range from characterizing regional differences in the overlap between communication and infection network layers, through understanding how our three mechanisms of learning contribute to individual behaviour over time, to extending beyond these to consider new learning mechanisms.

Social learning affects not just the public response [66] but also policy decisions [67], and models have explored the interaction between mitigation measures—such as hospitalizations, lockdowns or digital tracing—with infection rate [68–71]. Our work has demonstrated the critical and inextricably intertwined roles that social construction of risk perception, community awareness of disease incidence, and reassurance effects from local absence of active cases can have on the success of outbreak mitigative policies. We see footprints of these effects in the observed dynamics of the COVID-19 pandemic as we write, and we advocate for additional discussion building on these explicit insights to become a greater focus for both research and policy.

Data accessibility. See https://github.com/matthewsilk/CoupledDynamics NetworkPaper/.

Authors' contributions. M.J.S.: conceptualization, data curation, formal analysis, investigation, methodology, software, visualization, writing—original draft, writing—review and editing; S.C.: data curation, formal analysis, visualization, writing—review and editing; R.A.B.: supervision, visualization, writing—review and editing; N.H.F.: conceptualization, funding acquisition, methodology, supervision, writing—original draft, writing—review and editing. All authors gave final approval for publication and agreed to be held accountable for the work performed therein.

Competing interests. We declare we have no competing interests.

Funding. We gratefully acknowledge funding support for this work from NSF DEB 2028710.

Acknowledgements. We thank Mario Small for useful conversations in formulating these questions and for comments on the draft. We thank Dave Hodgson for allowing M.J.S. time to pursue this work.

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
