## [Peer Review File · Proceedings of the Royal Society B: Biological Sciences]

Review History

RSPB-2020-2739.R0 (Original submission)

Review form: Reviewer 1

Recommendation

Major revision is needed (please make suggestions in comments)

Scientific importance: Is the manuscript an original and important contribution to its field?

Excellent

General interest: Is the paper of sufficient general interest?

Excellent

Quality of the paper: Is the overall quality of the paper suitable?

Good

Is the length of the paper justified?

Yes

Should the paper be seen by a specialist statistical reviewer?

No

Do you have any concerns about statistical analyses in this paper? If so, please specify them explicitly in your report.

No

It is a condition of publication that authors make their supporting data, code and materials available - either as supplementary material or hosted in an external repository. Please rate, if applicable, the supporting data on the following criteria.

Is it accessible?

Yes

Is it clear?

Yes

Is it adequate?

Yes

Do you have any ethical concerns with this paper?

No

Comments to the Author

In this paper the authors develop a multiplex network model with infection processes in one layer and social process in the other layer. They explore the role of social forces and awareness of epidemic processes in network neighbours on the course of an epidemic in the context of COVID-19. They find that the timing of introducing interventions is crucial. If introduced too early, normalisation of perceived infection risk in a low-incidence environment may cause rejection of protective behaviours later, when incidence becomes significant. As a result, there is a crucial time window which they term the "behaviourally receptive phase", where the uptake of measures can be maximized. The authors also explore the impact of modularity and homophily on model dynamics.

This paper introduces a high-quality theoretical model that generates new hypotheses about optimizing control policy in the light of behavioural feedbacks. The model is adequately documented and parameterized. I think the paper will be of significant interest to individuals working in infectious disease modelling and is a great fit to Proceedings B. My comments pertain only to the presentation and integration with existing literature:

Major comments:

1. The authors should better contextualize their findings in the COVID-19 pandemic literature and the social psychology of infectious diseases literature. The social and psychological justification for their assumptions, such as through the Health Beliefs Model, could be better developed in the Introduction section. And in the Discussion section, they could comment with specific reference to the social science and epidemiological work on COVID-19 epidemics in various populations. For instance, this paper may be relevant to the question of when to introduce restrictions (https://papers.ssrn.com/sol3/papers.cfm?abstract_id=3575004). At the moment, the Discussion is heavily under-referenced. Better contextualization and justification are particularly important given the implications of a hypothesis that says we should wait until the time is right to impose restrictions, when the public health mantra for decades has been "act soon, act decisively".
2. The discussion is lengthy and should be shortened to make space for a paragraph on model limitations and their impact on the model predictions. This again pertains to my point above about implications of research suggesting that it is possible to implement public health interventions too quickly. On this particular note, could the authors discuss how this approach might be applied in practice?

3. The presentation of the results relies on the severity index. This is a useful metric, but it would be helpful to see some more figures in the main text that show results such as time series and epidemic final sizes, to make the results more approachable to a general biological sciences reader.

Minor comments:

1. SI appendix lines 685-66, "Parameter values were selected after a preliminary exploration of wider parameter space" – please provide more information on how the parameter values were selected.
2. Line 714 and Table S6: parameter values for incubation period need literature sources
3. The way that the results section is written is quite technical. The authors should make revisions to make it more accessible to a non-expert audience.

Review form: Reviewer 2

Recommendation

Major revision is needed (please make suggestions in comments)

Scientific importance: Is the manuscript an original and important contribution to its field?

Good

General interest: Is the paper of sufficient general interest?

Good

Quality of the paper: Is the overall quality of the paper suitable?

Marginal

Is the length of the paper justified?

Yes

Should the paper be seen by a specialist statistical reviewer?

No

Do you have any concerns about statistical analyses in this paper? If so, please specify them explicitly in your report.

No

It is a condition of publication that authors make their supporting data, code and materials available - either as supplementary material or hosted in an external repository. Please rate, if applicable, the supporting data on the following criteria.

Is it accessible?

Yes

Is it clear?

No

Is it adequate?

No

Do you have any ethical concerns with this paper?

No

Comments to the Author

This manuscript considers the coupled dynamics of disease transmission and spread of concern. A simulation model is constructed and run over a several parameterizations. The level of outbreak severity is assessed.

Major Comments:

The methodology and results are not clearly explained. All supplemental figures are missing so it was impossible to evaluate multiple statements.

Methodology:

The methodology is difficult to follow in the text as it is split between the main text and supplement. A table summarizing the different conditions, starting values, etc would make following the multiple conditions easier.

- How are concern and adherence tracked through the simulation? How does concern change?
- Individuals appear to be infectious for an average of 20 days (6.7 in I1, 10 in I2, 4.2 in I3), which is longer than most estimates.
- Five replicates per condition is low.

Line 141: What ages do children, young adult and old adult refer to?

Line 144: What do predisposition A and B refer to?

Line 174: Do individuals cut all connections when adherent (including child/parent)? It seems unrealistic that child-parent links would be cut.

Line 202-203: Outbreak severity is a major outcome considered. What is meant by this metric should be defined in the main text.

Line 214: What does "learning" mean in this context? It's not defined in the manuscript.

However, it is referred to in the manuscript (for example, line 226).

Line 679/686: Unclear what is meant by "Parameter values were selected after a preliminary exploration of wider parameter space." Are the values presented before or after that wider exploration?

Line 712: What is the transmission probability? It is not found in Table S6.

Line 719: How were these probabilities "adapted"?

Line 740: How were initial values of concern determined? What were the pre-defined starting values?

Assumptions:

Line 115: Justification for the assumption that social influence can elevate concern but not diminish it is needed.

Line 160: Justification for the assumption that predisposition in communities should be the same as at the overall population is needed

Line 301: It is mentioned that the US is a patchwork of town and cities of different population sizes. Why choose equal community sizes in the model?

Line 698: Justification that 50% probability to cut connections reflect a reasonable approximation of real-life is needed.

Results:

Line 230-231: How does Figure 3 represent time to reach the peak?

Line 334: How would one go about measuring community-level concern in order to pulse strategies?

Figure 3/Figure 5: What is "epidemic severity," the y-axis of the plots? Is this identical to outbreak severity?

Figure 4: It is unclear what is being plotted. How are "lines calculated from point estimates for the random intercept and random slope?"

Minor Comments:

Line 98: Should read “(the tendency to be affiliated ...”

Line 104: Does demography only refer to age groups or also to aging/birth/ death? If the former, it would be clearer to refer to “simplified age structure.” If the latter, these need to be defined.

Line 171: “Initial levels of concern” correspond to 50%, 20%, or 5% chance of adherence, not that they result in.

Line 290-291: Given that this is just one assumption on how concern about risk could occur, it is misleading to say that concern about risk between communities “will” clearly amplify patchiness. It would be more accurate to say that it “can” amplify.

Line 342: “deliver” should be “delivery.”

Line 342-346: Current state of vaccination should be updated given current events.

Line 694: Is the concern or adherence of the child identical to their parents?

Line 714: Incubation period is the time frame until onset of symptoms. It appears what is described is the latent period, i.e. the time until infectiousness.

Decision letter (RSPB-2020-2739.R0)

21-Jan-2021

Dear Professor Fefferman:

I am writing to inform you that your manuscript RSPB-2020-2739 entitled "Improving pandemic mitigation policies across communities through coupled dynamics of risk perception and infection" has, in its current form, been rejected for publication in Proceedings B.

This action has been taken on the advice of referees, who have recommended that substantial revisions are necessary. With this in mind we would be happy to consider a resubmission, provided the comments of the referees are fully addressed. However please note that this is not a provisional acceptance.

Sincerely,
 Professor Hans Heesterbeek
 mailto: proceedingsb@royalsociety.org

Associate Editor

Board Member: 1

Comments to Author:

Both referees find a great deal of promise in this paper and I agree. It presents an analysis of an interesting model of disease transmission, behaviour, and the interaction between the two. This is clearly very topical and I think the paper will be of broad interest. I agree with the referees, however, that more could be done to integrate the study into the literature. Some of the methodology could also be explain more clearly, as suggested by one of the referees.

Reviewer(s)' Comments to Author:

Referee: 1

Comments to the Author(s)

In this paper the authors develop a multiplex network model with infection processes in one layer and social process in the other layer. They explore the role of social forces and awareness of epidemic processes in network neighbours on the course of an epidemic in the context of COVID-19. They find that the timing of introducing interventions is crucial. If introduced too early, normalisation of perceived infection risk in a low-incidence environment may cause rejection of protective behaviours later, when incidence becomes significant. As a result, there is a crucial time window which they term the "behaviourally receptive phase", where the uptake of measures can be maximized. The authors also explore the impact of modularity and homophily on model dynamics.

This paper introduces a high-quality theoretical model that generates new hypotheses about optimizing control policy in the light of behavioural feedbacks. The model is adequately documented and parameterized. I think the paper will be of significant interest to individuals working in infectious disease modelling and is a great fit to Proceedings B. My comments pertain only to the presentation and integration with existing literature:

Major comments:

1. The authors should better contextualize their findings in the COVID-19 pandemic literature and the social psychology of infectious diseases literature. The social and psychological justification for their assumptions, such as through the Health Beliefs Model, could be better developed in the Introduction section. And in the Discussion section, they could comment with specific reference to the social science and epidemiological work on COVID-19 epidemics in various populations. For instance, this paper may be relevant to the question of when to introduce restrictions (https://papers.ssrn.com/sol3/papers.cfm?abstract_id=3575004). At the moment, the Discussion is heavily under-referenced. Better contextualization and justification are particularly important given the implications of a hypothesis that says we should wait until the time is right to impose restrictions, when the public health mantra for decades has been "act soon, act decisively".
2. The discussion is lengthy and should be shortened to make space for a paragraph on model limitations and their impact on the model predictions. This again pertains to my point above about implications of research suggesting that it is possible to implement public health interventions too quickly. On this particular note, could the authors discuss how this approach might be applied in practice?
3. The presentation of the results relies on the severity index. This is a useful metric, but it would be helpful to see some more figures in the main text that show results such as time series and epidemic final sizes, to make the results more approachable to a general biological sciences reader.

Minor comments:

1. SI appendix lines 685-66, "Parameter values were selected after a preliminary exploration of wider parameter space" – please provide more information on how the parameter values were selected.
2. Line 714 and Table S6: parameter values for incubation period need literature sources
3. The way that the results section is written is quite technical. The authors should make revisions to make it more accessible to a non-expert audience.

Referee: 2

Comments to the Author(s)

This manuscript considers the coupled dynamics of disease transmission and spread of concern. A simulation model is constructed and run over a several parameterizations. The level of outbreak severity is assessed.

Major Comments:

The methodology and results are not clearly explained. All supplemental figures are missing so it was impossible to evaluate multiple statements.

Methodology:

The methodology is difficult to follow in the text as it is split between the main text and supplement. A table summarizing the different conditions, starting values, etc would make following the multiple conditions easier.

- How are concern and adherence tracked through the simulation? How does concern change?
- Individuals appear to be infectious for an average of 20 days (6.7 in I1, 10 in I2, 4.2 in I3), which is longer than most estimates.
- Five replicates per condition is low.

Line 141: What ages do children, young adult and old adult refer to?

Line 144: What do predisposition A and B refer to?

Line 174: Do individuals cut all connections when adherent (including child/parent)? It seems unrealistic that child-parent links would be cut.

Line 202-203: Outbreak severity is a major outcome considered. What is meant by this metric should be defined in the main text.

Line 214: What does "learning" mean in this context? It's not defined in the manuscript.

However, it is referred to in the manuscript (for example, line 226).

Line 679/686: Unclear what is meant by "Parameter values were selected after a preliminary exploration of wider parameter space." Are the values presented before or after that wider exploration?

Line 712: What is the transmission probability? It is not found in Table S6.

Line 719: How were these probabilities "adapted"?

Line 740: How were initial values of concern determined? What were the pre-defined starting values?

Assumptions:

Line 115: Justification for the assumption that social influence can elevate concern but not diminish it is needed.

Line 160: Justification for the assumption that predisposition in communities should be the same as at the overall population is needed

Line 301: It is mentioned that the US is a patchwork of town and cities of different population sizes. Why choose equal community sizes in the model?

Line 698: Justification that 50% probability to cut connections reflect a reasonable approximation of real-life is needed.

Results:

Line 230-231: How does Figure 3 represent time to reach the peak?

Line 334: How would one go about measuring community-level concern in order to pulse strategies?

Figure 3/Figure 5: What is “epidemic severity,” the y-axis of the plots? Is this identical to outbreak severity?

Figure 4: It is unclear what is being plotted. How are “lines calculated from point estimates for the random intercept and random slope?”

Minor Comments:

Line 98: Should read “(the tendency to be affiliated ...”

Line 104: Does demography only refer to age groups or also to aging/birth/death? If the former, it would be clearer to refer to “simplified age structure.” If the latter, these need to be defined.

Line 171: “Initial levels of concern” correspond to 50%, 20%, or 5% chance of adherence, not that they result in.

Line 290-291: Given that this is just one assumption on how concern about risk could occur, it is misleading to say that concern about risk between communities “will” clearly amplify patchiness. It would be more accurate to say that it “can” amplify.

Line 342: “deliver” should be “delivery.”

Line 342-346: Current state of vaccination should be updated given current events.

Line 694: Is the concern or adherence of the child identical to their parents?

Line 714: Incubation period is the time frame until onset of symptoms. It appears what is described is the latent period, i.e. the time until infectiousness.

Author's Response to Decision Letter for (RSPB-2020-2739.R0)

See Appendix A.

RSPB-2021-0834.R0

Review form: Reviewer 1

Recommendation

Accept as is

Scientific importance: Is the manuscript an original and important contribution to its field?

Excellent

General interest: Is the paper of sufficient general interest?

Excellent

Quality of the paper: Is the overall quality of the paper suitable?

Excellent

Is the length of the paper justified?

Yes

Should the paper be seen by a specialist statistical reviewer?

No

Do you have any concerns about statistical analyses in this paper? If so, please specify them explicitly in your report.

No

It is a condition of publication that authors make their supporting data, code and materials available - either as supplementary material or hosted in an external repository. Please rate, if applicable, the supporting data on the following criteria.

Is it accessible?

Yes

Is it clear?

Yes

Is it adequate?

Yes

Do you have any ethical concerns with this paper?

No

Comments to the Author

The authors have responded to my comments in a satisfactory way. Thank you for the enjoyable read.

Review form: Reviewer 2

Recommendation

Major revision is needed (please make suggestions in comments)

Scientific importance: Is the manuscript an original and important contribution to its field?

Good

General interest: Is the paper of sufficient general interest?

Good

Quality of the paper: Is the overall quality of the paper suitable?

Acceptable

Is the length of the paper justified?

Yes

Should the paper be seen by a specialist statistical reviewer?

No

Do you have any concerns about statistical analyses in this paper? If so, please specify them explicitly in your report.

Yes

It is a condition of publication that authors make their supporting data, code and materials available - either as supplementary material or hosted in an external repository. Please rate, if applicable, the supporting data on the following criteria.

Is it accessible?

Yes

Is it clear?

Yes

Is it adequate?

Yes

Do you have any ethical concerns with this paper?

No

Comments to the Author

The authors have significantly improved the clarity of the manuscript through the inclusion of definitions, consistency in language and more detail. However, several concerns remain.

Some of the methodology remains unclear:

- How probabilities are adapted remains unclear? The authors note "We made small post-hoc changes to these probabilities to fit outcomes of simulations to observed rates of hospitalisation and mortality when the models were developed. We now clarify this at L131-133." Line 131 - 133 states "We recorded epidemic outcomes for a range of network structures and proposed models for the change in concern of individuals over time." What are these post-hoc changes? What is the observed hospitalization and mortality used?
- If connections are not cut between adult and children, does that mean that if those individuals are in I2 or I3, they can transmit?
- There are a number of simplifying assumptions (e.g. that social influence can elevate concern but not lower it, that predisposition in communities is the same as the overall population, equal community size, fraction of contacts cut with concern). While these are now mentioned in the discussion, in the context that it would be nice to vary them in future work, there are no reasons given for the choices. Furthermore, the reason given fractions of contacts cut with concern compares to government enforced lockdowns, which is not the type of behavioral changes being proposed here.

Justification is lacking at a number of points:

- The justification for the length of the infectious period remains unclear. If individuals are truly not infectious during I2 and I3 due to cutting of contacts, they should be considered non-infectious. Otherwise, the underlying assumption of the model is that individuals can be infectious for 20+ days, which is beyond current estimates. The suggestion that this might represent variants still does not account for infectious periods of this length.
- The justification for the low number of replicates (5) is that there are clear distinctions between conditions. This does not appear evident from the figures presented. Many conditions seem to result in overlapping outcomes, given the variation in parameters, and it would be important to know if these outcomes are truly different.

Minor:

Figure 5 lacks a color scale.

The SI figures lack full information. There is no color scale for Figures S1, S2, S5. Figures S2 and S5 have no axes values.

Decision letter (RSPB-2021-0834.R0)

05-May-2021

Dear Professor Fefferman:

Your manuscript has now been peer reviewed and the reviews have been assessed by an Associate Editor. The reviewers' comments (not including confidential comments to the Editor) and the comments from the Associate Editor are included at the end of this email for your reference. As you will see, the reviewers and the Editors have raised some issues and we would like to invite you to revise your manuscript to address them.

Research ethics:

Use of animals and field studies:

It is a condition of publication that you make available the data and research materials supporting the results in the article (<https://royalsociety.org/journals/authors/author-guidelines/#data>). Datasets should be deposited in an appropriate publicly available repository and details of the associated accession number, link or DOI to the datasets must be included in the Data Accessibility section of the article (<https://royalsociety.org/journals/ethics-policies/data-sharing-mining/>). Reference(s) to datasets should also be included in the reference list of the article with DOIs (where available).

Please submit a copy of your revised paper within three weeks. If we do not hear from you within this time your manuscript will be rejected. If you are unable to meet this deadline please let us know as soon as possible, as we may be able to grant a short extension.

Best wishes,
Professor Hans Heesterbeek
mailto:proceedingsb@royalsociety.org

Associate Editor
Comments to Author:

I think the authors have done a great job in their revisions but, as one of the referees remarks, there are still some outstanding issues that need attention. In particular, I would like to see the authors provide more detail about the "post-hoc" modifications used since these kinds of tweaks always make me nervous. Also, a number of the assumptions could still use some justification. And I don't think the existence of variants can be used as justification for 20+ days since there is very little evidence that infectious period differs with variants. Finally, I think the referee has an important concern about the number of replicates presented.

Reviewer(s)' Comments to Author:

Referee: 1

Comments to the Author(s).

The authors have responded to my comments in a satisfactory way. Thank you for the enjoyable read.

Referee: 2

Comments to the Author(s).

The authors have significantly improved the clarity of the manuscript through the inclusion of definitions, consistency in language and more detail. However, several concerns remain.

Some of the methodology remains unclear:

- How probabilities are adapted remains unclear? The authors note “We made small post-hoc changes to these probabilities to fit outcomes of simulations to observed rates of hospitalisation and mortality when the models were developed. We now clarify this at L131-133.” Line 131 – 133 states “We recorded epidemic outcomes for a range of network structures and proposed models for the change in concern of individuals over time.” What are these post-hoc changes? What is the observed hospitalization and mortality used?
- If connections are not cut between adult and children, does that mean that if those individuals are in I2 or I3, they can transmit?
- There are a number of simplifying assumptions (e.g. that social influence can elevate concern but not lower it, that predisposition in communities is the same as the overall population, equal community size, fraction of contacts cut with concern). While these are now mentioned in the discussion, in the context that it would be nice to vary them in future work, there are no reasons given for the choices. Furthermore, the reason given fractions of contacts cut with concern compares to government enforced lockdowns, which is not the type of behavioral changes being proposed here.

Justification is lacking at a number of points:

- The justification for the length of the infectious period remains unclear. If individuals are truly not infectious during I2 and I3 due to cutting of contacts, they should be considered non-infectious. Otherwise, the underlying assumption of the model is that individuals can be infectious for 20+ days, which is beyond current estimates. The suggestion that this might represent variants still does not account for infectious periods of this length.
- The justification for the low number of replicates (5) is that there are clear distinctions between conditions. This does not appear evident from the figures presented. Many conditions seem to result in overlapping outcomes, given the variation in parameters, and it would be important to know if these outcomes are truly different.

Minor:

Figure 5 lacks a color scale.

The SI figures lack full information. There is no color scale for Figures S1, S2, S5. Figures S2 and S5 have no axes values.

Author's Response to Decision Letter for (RSPB-2021-0834.R0)

See Appendix B.

Decision letter (RSPB-2021-0834.R1)

23-Jun-2021

Dear Professor Fefferman

I am pleased to inform you that your manuscript entitled "Improving pandemic mitigation policies across communities through coupled dynamics of risk perception and infection" has been accepted for publication in Proceedings B.

COVID-19 rapid publication process: We are taking steps to expedite the publication of research relevant to the pandemic. If you wish, you can opt to have your paper published as soon as it is ready, rather than waiting for it to be published the following Wednesday.

This means your paper will not be included in the weekly media round-up which the Society sends to journalists ahead of publication. However, it will appear in the COVID-19 Publishing Collection which journalists will be directed to each week

(<https://royalsocietypublishing.org/topic/special-collections/novel-coronavirus-outbreak>)

If you wish to have your paper published immediately please notify production@royalsociety.org and press@royalsociety.org when you respond to this email.

Data Accessibility section

Open Access

Paper charges

Sincerely,

Professor Hans Heesterbeek

Associate Editor:

Board Member

Comments to Author:

(There are no comments.)

Appendix A

Dear Prof. Heesterbeek,

Please find attached the revision of our manuscript, RSPB-2020-2739, "Improving pandemic mitigation policies across communities through coupled dynamics of risk perception and infection."

We are very grateful for the helpful comments from the Associate Editor and the Reviewers. We believe the results we present will continue to be relevant to policy makers during this current phase of the global pandemic in which vaccines and variants are still subject to behavior governed by the dynamic coupling of risk perception and epidemic spread. As you will see, we have substantially revised the text and have re-worked many of the figures to address the suggestions made. We believe that the changes we have made in response to the comments have helped to greatly strengthen the manuscript.

We are hopeful that our paper may now be suitable for publication in *the Proceedings of the Royal Society of London, B*.

Thank you,

Matthew J Silk, Simon Carrignon, R. Alexander Bentley, and Nina H Fefferman

Associate Editor

Board Member: 1

Comments to Author:

Both referees find a great deal of promise in this paper and I agree. It presents an analysis of an interesting model of disease transmission, behaviour, and the interaction between the two. This is clearly very topical and I think the paper will be of broad interest. I agree with the referees, however, that more could be done to integrate the study into the literature. Some of the methodology could also be explain more clearly, as suggested by one of the referees.

Response: We thank the associate editor and both reviewers for their kind and constructive comments. The revised version of the paper is greatly improved as an outcome of the suggestions made. In our responses we have explained why we feel it is appropriate to revise the paper without re-running the simulations but can do this if required. We feel it is highly unlikely that making some of the small adjustments suggested would meaningfully change our results, and some of the additional ideas suggested fall outside the scope of this model given the extent of the parameter space we are already exploring (although they reflect things we are interested in and keen to explore in future).

Line numbers refer to the clean (no tracked changes) version of the revision. We provide separate line numbers for the main text (LX) and the supplementary (SI LX).

Reviewer(s)' Comments to Author:

Referee: 1

Comments to the Author(s)

In this paper the authors develop a multiplex network model with infection processes in one layer and social process in the other layer. They explore the role of social forces and awareness of epidemic processes in network neighbours on the course of an epidemic in the context of COVID-19. They find that the timing of introducing interventions is crucial. If introduced too early, normalisation of perceived infection risk in a low-incidence environment may cause rejection of protective behaviours later, when incidence becomes significant. As a result, there is a crucial time window which they term the “behaviourally receptive phase”, where the uptake of measures can be maximized. The authors also explore the impact of modularity and homophily on model dynamics.

This paper introduces a high-quality theoretical model that generates new hypotheses about optimizing control policy in the light of behavioural feedbacks. The model is adequately documented and parameterized. I think the paper will be of significant interest to individuals working in infectious disease modelling and is a great fit to Proceedings B. My comments pertain only to the presentation and integration with existing literature:

Response: We thank the reviewer for their kind comments on the modelling framework and their thoughtful and helpful comments with the presentation of the paper. We have outlined how we have addressed these comments in detail below. Overall, we have shortened the discussion and edited it to place our study within the context of the literature. We have also edited the methods and results based on comments of both reviewers to make them more fully explained and accessible.

Major comments:

1. The authors should better contextualize their findings in the COVID-19 pandemic literature and the social psychology of infectious diseases literature. The social and psychological justification for their assumptions, such as through the Health Beliefs Model, could be better developed in the Introduction section. And in the Discussion section, they could comment with specific reference to the social science and epidemiological work on COVID-19 epidemics in various populations. For instance, this paper may be relevant to the question of when to introduce restrictions (https://papers.ssrn.com/sol3/papers.cfm?abstract_id=3575004). At the moment, the Discussion is heavily under-referenced. Better contextualization and justification are particularly important given the implications of a hypothesis that says we should wait until the time is right to impose restrictions, when the public health mantra for decades has been “act soon, act decisively”.

Response: We have edited the discussion section to provide better context to our results and provide more citations. This includes more reference to other covid modelling studies (many of which have appeared very recently in the literature!). We have also adjusted our main message in the paper to make it clearer (in terms of encouraging a strong response).

2. The discussion is lengthy and should be shortened to make space for a paragraph on model limitations and their impact on the model predictions. This again pertains to my point above about implications of research suggesting that it is possible to implement public health interventions too quickly. On this particular note, could the authors discuss how this approach might be applied in practice?

Response: We have shortened multiple sections of the discussion (with the exception of the additions outlined in our previous comment). We have also added a paragraph specifically highlighting some of the limitations of our modelling framework and some untested parameters (some of which were picked up on by reviewer 2) [L361-384]. We have made our description of how this approach could be applied in practice more explicit (e.g. L332-335, L341-342 and L354-360).

3. The presentation of the results relies on the severity index. This is a useful metric, but it would be helpful to see some more figures in the main text that show results such as time series and epidemic final sizes, to make the results more approachable to a general biological sciences reader.

Response: We have made various changes to the results to make them more accessible to the reader (also suggested by reviewer 2). We have added an additional figure (now Fig. 3) showing a selection of epidemic curves and relating them to our measure of epidemic severity.

Minor comments:

1. SI appendix lines 685-66, “Parameter values were selected after a preliminary exploration of wider parameter space”—please provide more information on how the parameter values were selected.

Response: We have provided a clearer description of how parameter values were selected (SI L82-84). In brief we conducted initial runs with a wider distribution of the parameters of greatest interest (the three learning effects) and selected values from the range over which changes in these parameters produced meaningfully different outcomes. The values of these parameters all fall within a reasonable expectation of how people could respond/ behave in reality.

2. Line 714 and Table S6: parameter values for incubation period need literature sources

Response: All parameter values were taken directly from or implemented to be quantitatively similar to the paper mentioned in the first line of this section. We have now amended this to “We modelled the spread of SARS-CoV-2 using a stochastic SEIRD model adapted from (8), with parameter values based on this model except when stated.” for clarity (SI L117-118).

3. The way that the results section is written is quite technical. The authors should make revisions to make it more accessible to a non-expert audience.

Response: We have edited the text in the results section to make it more accessible to a general reader and also provided the extra figure as suggested above.

Referee: 2

Comments to the Author(s)

This manuscript considers the coupled dynamics of disease transmission and spread of concern. A simulation model is constructed and run over a several parameterizations. The level of outbreak severity is assessed.

Major Comments:

The methodology and results are not clearly explained. All supplemental figures are missing so it was impossible to evaluate multiple statements.

Response: We have now made extensive revisions to the methods and supplementary materials to improve their clarity. We have also ensured the supplementary figures are included in the supplementary materials (within the PDF provided).

Methodology:

The methodology is difficult to follow in the text as it is split between the main text and supplement. A table summarizing the different conditions, starting values, etc would make following the multiple conditions easier.

Response: We have included a depiction of the conditions used in the simulation in the main text as suggested (Fig. 2b) - we decided to include it in this way to save space and link it explicitly to the flow chart of the model.

- How are concern and adherence tracked through the simulation? How does concern change?

Response: We track the concern and adherence of all individuals throughout the simulations and have changed the methods/supplementary material to make this clear (L171). We record summary data on the proportion of adherent individuals within each community to avoid huge output files.

- Individuals appear to be infectious for an average of 20 days (6.7 in I1, 10 in I2, 4.2 in I3), which is longer than most estimates.

Response: Our response here is complex, sorry! We agree with the reviewer about the long symptomatic period, but for various reasons think that the approximations we made are reasonable because of the mechanics of our model. We have tried to explain these below and have also better explained our rationale in the paper (L186-188 and SI L138-141). We do not feel that making small adjustments to the mechanics described below would qualitatively change the results.

Because individuals can transition from I2->I3 (effectively be hospitalised) at any point while in the I2 category the period of that individuals show symptoms in our models has a mean of ~17.2 days, median of 17 days and mode of 15-17 days (varying due to the stochastic nature of the process). Individuals in I2 and I3 are rewired to have minimal/negligible connection strengths in the infection layer of the multiplex network and so individuals in the model are only “meaningfully” infectious for the I1 period (mean of 6.7 days). We originally based this parameter value on information from Lofgren et al. (2020) when much more limited information was available. While it is longer than the pre-symptomatic period, in our model it will (effectively) reflect both the pre-symptomatic period and the period prior to a person testing positive or independently realising they have COVID-19 and therefore a sensible approximation.

Another relevant point is that there is evidence for some of the current variants of concern being more transmissible in part because of a longer infectious period than previous variants. While not related to our original decision-making, it shows that the values selected are reasonable and appropriate.

- Five replicates per condition is low.

Response: Given the number of parameters varied and the approach used to present and summarise the patterns in the results, we feel that using 5 replicates of each scenario is sufficient. Given the output shows clear patterns and differences between different scenarios, we think adding more replicates will only reproduce the already observed trends and so be of limited benefit relative to the computational/energy costs. However, we can run further replicates if it is still considered necessary.

Line 141: What ages do children, young adult and old adult refer to?

Response: Children are 18 and under, old adult 65 and over. This is now clarified in the text (L123-125)

Line 144: What do predisposition A and B refer to?

Response: We use the predispositions in this model to represent any situation where there may be homophily in who people communicate or come into physical contact with. A possible example would be partisanship in the USA. We now provide this example in the text to better explain the idea (L125-128).

Line 174: Do individuals cut all connections when adherent (including child/parent)? It seems unrealistic that child-parent links would be cut.

Response: Connections within families are not cut. We apologise for not describing this previously. We now mention this briefly in the methods (L174) and a more detailed description of the rewiring algorithm is now provided in the supplementary material (SI L106-108).

Line 202-203: Outbreak severity is a major outcome considered. What is meant by this metric should be defined in the main text.

Response: Epidemic severity is now defined in the main text (L203-211) and we use Figure 3 (newly added) to illustrate the concept.

Line 214: What does “learning” mean in this context? It’s not defined in the manuscript. However, it is referred to in the manuscript (for example, line 226).

Response: We have now provided a definition of what we mean by learning (L219-220).

Line 679/686: Unclear what is meant by “Parameter values were selected after a preliminary exploration of wider parameter space.” Are the values presented before or after that wider exploration?

Response: We have provided a clearer description of how parameter values were selected (SI L82-84). In brief we conducted initial runs with a wider distribution of the parameters of greatest interest (the three learning effects) and selected values from the range over which changes in these parameters produced meaningfully different outcomes. The values of these parameters all fall within a reasonable expectation of how people could respond/ behave in reality.

Line 712: What is the transmission probability? It is not found in Table S6.

Response: Table S6 included “Probability of infection per contact”, we have now changed this to “Transmission probability per contact”

Line 719: How were these probabilities “adapted”?

Response: We made small post-hoc changes to these probabilities to fit outcomes of simulations to observed rates of hospitalisation and mortality when the models were developed. We now clarify this at L131-133.

Line 740: How were initial values of concern determined? What were the pre-defined starting values?

Response: The pre-defined starting values for individual concern were set so that the expected number of adherent individuals matched the initial levels of adherence in each community (5%, 20% and 50% adherent) [see L168-169 and SI L71-73].

Assumptions:

Line 115: Justification for the assumption that social influence can elevate concern but not diminish it is needed.

Response: This is something we thought carefully about when designing the modelling framework used in the study. Our models have the capacity to do this but we decided not to do it for this study as allowing social influence to elevate but not reduce concern is conservative in respect to one of the primary focuses of the study, the reassurance effect. By not allowing social influence to reduce concern we can be clear just how important this reassurance effect can be. We agree that it is something of great interest in general in these types of models and have added text in the discussion to talk about the possible implications and the need for future work to explore this (new paragraph at L361-384).

Line 160: Justification for the assumption that predisposition in communities should be the same as at the overall population is needed

Response: Predispositions in communities being the same as the overall population is again a simplifying assumption that is conservative with respect to our outcomes of interest. Once again we agree completely that it would be incredibly interesting to investigate the impacts of communities varying in their initial levels of concern/adherence but this is one for future work to address, we have briefly mentioned this now in the discussion (new paragraph at L361-384).

Line 301: It is mentioned that the US is a patchwork of town and cities of different population sizes. Why choose equal community sizes in the model?

Response: Similar to the previous two responses, we absolutely agree that this is an interesting question and worthy of further research. We anticipate that our work can be built on by us and others to explore such questions. We have briefly mentioned this in the discussion alongside the previous point made (new paragraph at L361-384) when discussing model limitations/extensions.

Line 698: Justification that 50% probability to cut connections reflect a reasonable approximation of real-life is needed.

Response: We have added the text “The proportion of connections cut in our model is lower than the 70-75% reduction in contacts recorded during government enforced lockdowns in Europe, and therefore likely to represent a reasonable approximation to real-world changes” (SI L107-110). Incorporating a greater reduction in contacts would likely improve the effectiveness with which communities flattened the infection curve, although we would expect our results to be qualitatively the same with changes in this value.” and included appropriate citations to these statements.

Results:

Line 230-231: How does Figure 3 represent time to reach the peak?

Response: This was an error in our writing and has now been corrected to refer to the correct figure in the supplementary material.

Line 334: How would one go about measuring community-level concern in order to pulse strategies?

Response: There are many ways to estimate community-level concern. For example, community concern can be estimated indirectly by comparing the magnitude of effort, information, or investment required to shift community behavior. There are also direct survey methods which have been employed successfully to understand community concern about both individual and communal risk (ranging from health effects of pesticide application to the building of cell phone towers). These methods have been best developed in the context of risks from natural hazards (see Wachinger, Gisela, Ortwin Renn, Chloe Begg, and Christian Kuhlicke. "The risk perception paradox—implications for governance and communication of natural hazards." *Risk analysis* 33, no. 6 (2013): 1049-1065.) but have been studied to produce a general understanding (see Fitchen, Janet M., Jenifer S. Heath, and June Fessenden-Raden. "Risk perception in community context: a case study." In *The social and cultural construction of risk*, pp. 31-54. Springer, Dordrecht, 1987.; Barnes, Paul. "Approaches to community safety: risk perception and social meaning." *Australian Journal of Emergency Management* 17, no. 1 (2002): 15-23.) We have now added reference to these efforts in the main text (L382-383), although we do not discuss them explicitly as we believe the greatest use is simply in the reader knowing such methods exist and have been usefully employed.

Figure 3/Figure 5: What is “epidemic severity,” the y-axis of the plots? Is this identical to outbreak severity?

Response: We now use epidemic severity throughout for consistency (we have chosen arbitrarily and can change if preferred).

Figure 4: It is unclear what is being plotted. How are “lines calculated from point estimates for the random intercept and random slope?”

Response: We have revised the caption to better explain what was done here.

Minor Comments:

Line 98: Should read “(the tendency to be affiliated ...”

Response: Changed as suggested.

Line 104: Does demography only refer to age groups or also to aging/birth/death? If the former, it would be clearer to refer to “simplified age structure.” If the latter, these need to be defined.

Response: We have changed to age structure as suggested (L104)

Line 171: “Initial levels of concern” correspond to 50%, 20%, or 5% chance of adherence, not that they result in.

Response: Changed as suggested (L168)

Line 290-291: Given that this is just one assumption on how concern about risk could occur, it is misleading to say that concern about risk between communities “will” clearly amplify patchiness. It would be more accurate to say that it “can” amplify.

Response: This language has been removed

Line 342: “deliver” should be “delivery.”

Response: This language has been removed with other revisions

Line 342-346: Current state of vaccination should be updated given current events.

Response: We have now updated the language to reflect the ongoing roll-out of vaccine options.

Line 694: Is the concern or adherence of the child identical to their parents?

Response: We have now corrected this sentence (SI L100-101)

Line 714: Incubation period is the time frame until onset of symptoms. It appears what is described is the latent period, i.e. the time until infectiousness.

Response: Changed to latent period as suggested (SI L126 and Table S6).

Appendix B

Dear Prof Heesterbeek,

Please find attached our attached our revised manuscript "*Improving pandemic mitigation policies across communities through coupled dynamics of risk perception and infection*" for your consideration. Based on comments from the associate editor and second reviewer we re-parametrised and re-ran our models to deal with their most substantial concerns with only small changes in the qualitative results (predominantly reduced or slightly changed patterns between the different networks). Once again, the other comments have been helpful in improving the clarity of the manuscript. We hope that you find the revised manuscript suitable for publication in the journal.

As requested, versions of the main manuscript and supplementary information with changes since the last version tracked are appended to this response.

Thank you for your time. We look forward to hearing from you.

Yours sincerely,
Nina Fefferman

Associate Editor

Comments to Author:

I think the authors have done a great job in their revisions but, as one of the referees remarks, there are still some outstanding issues that need attention. In particular, I would like to see the authors provide more detail about the "post-hoc" modifications used since these kinds of tweaks always make me nervous. Also, a number of the assumptions could still use some justification. And I don't think the existence of variants can be used as justification for 20+ days since there is very little evidence that infectious period differs with variants. Finally, I think the referee has an important concern about the number of replicates presented.

Response: We thank the associate editor and reviewer for their constructive comments again. Given the remaining concerns we have re-run the model with new parameters for the disease model (apologies for the delay this caused in resubmitting the revised manuscript) and better explained how some of these parameters were selected (after re-parameterisation). We have also added further replicates and have demonstrated at the bottom of the response that this number of replicates is sufficient). We would also highlight that our analytical approach draws information from multiple runs of the

simulation meaning that our power to make inferences about importance of different effects is based on more than the number of replicates of each unique condition.

Reviewer(s)' Comments to Author:

Referee: 1

Comments to the Author(s).

The authors have responded to my comments in a satisfactory way. Thank you for the enjoyable read.

Response: We thank the reviewer for their previous comments that greatly improved the manuscript

Referee: 2

Comments to the Author(s).

The authors have significantly improved the clarity of the manuscript through the inclusion of definitions, consistency in language and more detail. However, several concerns remain.

Response: We thank the reviewer for distilling their remaining concerns which we feel we have fully addressed through a combination of re-running re-parameterised models, and through further changes to the text.

Some of the methodology remains unclear:

- How probabilities are adapted remains unclear? The authors note “We made small post-hoc changes to these probabilities to fit outcomes of simulations to observed rates of hospitalisation and mortality when the models were developed. We now clarify this at L131-133.” Line 131 – 133 states “We recorded epidemic outcomes for a range of network structures and proposed models for the change in concern of individuals over time.” What are these post-hoc changes? What is the observed hospitalization and mortality used?

Response: We have re-parameterised the models and have re-calculated some of these parameters. We now more clearly explain how we selected these values (Supplementary L150-154). In brief, we set up a negative control run in which we ran simply the epidemiological component of the model (i.e. zero concern) in network 1. We selected a value of the transmission probability that resulted in ~80% of the population being infected without any behavioural change (as explained previously). We then adjusted probabilities of hospitalisation and death to produce numbers broadly representative of real-world data. We explain this process in much more depth in the supplement and

provide links to the data sources used. Stochasticity (we have low number of deaths and hospitalisations in each simulation run) means that we have focussed on generating a broadly representative scenario rather than exact match. [We would also highlight that due to the stochasticity around hospitalisations and deaths caused by our relatively small total population size, we do not in fact use information on hospitalisation and death in this study].

- If connections are not cut between adult and children, does that mean that if those individuals are in I2 or I3, they can transmit?

Response: Connections among adults and children are maintained when behavioural changes are caused by concern/adherence but not when changes are caused by illness. Therefore there is a very low chance of transmission in these scenarios. We have now clarified this in the text (L194 and Supplementary L160-161).

- There are a number of simplifying assumptions (e.g. that social influence can elevate concern but not lower it, that predisposition in communities is the same as the overall population, equal community size, fraction of contacts cut with concern). While these are now mentioned in the discussion, in the context that it would be nice to vary them in future work, there are no reasons given for the choices. Furthermore, the reason given fractions of contacts cut with concern compares to government enforced lockdowns, which is not the type of behavioral changes being proposed here.

Response: We have revisited some of these issues and explained in more detail why we made these specific choices (L147-149, L159-161, L173-177, L181-183). To some extent, some of these choices represent arbitrary decisions in which choices were made that were realistic and reasonable to enable us to concentrate on specific regions of parameter space. Making these choices are necessary in limiting the parameter space to be explored and the extent of the results that need to be explained. We chose the simplifying assumption that social influence can only increase concern as it is conservative with regards to the impact of reassurance (a major focus of the paper) and have clarified this (L173-177). We are working on the impact of changing this assumption currently. Homogeneity in predispositions and community sizes are logical simplifying assumptions to make to test general theories about learning and help capture when it is important. We absolutely agree that varying these parameters is interesting, and will be achievable with the sorts of data collected during the pandemic if applied to this or a similar modelling framework. In terms of the justification using government enforced interventions, we had not intended to make these as a comparison per se, more as an indication of the likely upper bound for a reduction in contacts, demonstrating that our selected parameter choice falls within a reasonable region,

which we have now clarified (Supplementary L115-121). There is likely considerable variation in the ability and desire to reduce social connections both within and between different communities and so we feel that picking a value below that achieved by government enforced “lockdowns” is a reasonable choice here (some people will achieve similar reductions in contacts in the absence of these lockdowns, e.g.

<https://www.nature.com/articles/s41467-021-20990-2>, while others will be able to reduce their contacts by much less).

Justification is lacking at a number of points:

- The justification for the length of the infectious period remains unclear. If individuals are truly not infectious during I2 and I3 due to cutting of contacts, they should be considered non-infectious. Otherwise, the underlying assumption of the model is that individuals can be infectious for 20+ days, which is beyond current estimates. The suggestion that this might represent variants still does not account for infectious periods of this length.

Response: We have re-parameterised the model to change the course of disease given the continued concerns this may influence the results. This has resulted in a shorter timeframe of infection (mean durations: 4 days incubation, 4 days pre/mildly symptomatic and ~8 days ill unless infection progresses). We have chosen to stick with our choice of modelling disease progression given there is a low chance of transmission from I2/I3 states and this may reflect reality.

- The justification for the low number of replicates (5) is that there are clear distinctions between conditions. This does not appear evident from the figures presented. Many conditions seem to result in overlapping outcomes, given the variation in parameters, and it would be important to know if these outcomes are truly different.

Response: We have increased the number of replicates per parameter combination from 5 to 10. Appended to the end of the response we demonstrate that 10 replicates is sufficient. We would also highlight our use of statistical models to describe patterns in the results draws inference from multiple unique sets of parameter values for each estimate.

Minor:

Figure 5 lacks a color scale.

Response: We have added a colour scale as suggested

The SI figures lack full information. There is no color scale for Figures S1, S2, S5. Figures S2 and S5 have no axes values.

Response: We have now tidied up and improved the figures in the supplement.

Number of replicates

We have increased the number of replicates from 5 to 10. Because we simulated a unique value of the Reassurance Effect for every simulation it is more involved to demonstrate when a sufficient number of replicates have been reached but we provide one such approach here.

Here we use a bootstrapping approach on replicate number (an identifier between 1 and 10). For the value i in the range 1 through 10 we sample with replacement i replicates 100 times.

e.g. when $i=2$ we could sample [1,3], [5,8] or [9,9] or when $i=6$ we could sample [1,4,4,7,8,10].

We then reconstruct the dataset using only the replicates in that bootstrap sampled set.

For each run we calculate the mean and standard deviation in the severity measure (a focus of our analysis of simulation results) for each combination of Network ID and Parameter ID (this is the combination that in the full analysis is replicated 10 times) for the dataset where initial concern is high and moderate (i.e. the two datasets that formed the focus of the analysis).

We then calculated the population-level variance in these within-group means and standard deviations and plotted them in the boxplots provided below.

As you can see from the boxplots, the population-variance in within-group means stabilizes from around 5/6 replicates and higher, and a similar pattern is apparent for within-group variances. This pattern is repeated (general pattern very similar indeed) when we consider datasets including runs with a high level of initial concern/adherence and a moderate level of initial concern/adherence.

Fig. R1. Population-level variance in within-group means as calculated using a bootstrapping sampling approach for the final dataset with high levels of initial concern/adherence. Variance stabilizes at between 5 and 7 replicates so that no meaningful gains are made when sampling more replicates than this.

Fig. R2. Population-level variance in within-group means as calculated using a bootstrapping sampling approach for the final dataset with high levels of initial concern/adherence. Variance stabilizes at ~5 replicates so that no meaningful gains are made when sampling more replicates than this.

Fig R3. Population-level variance in within-group means as calculated using a bootstrapping sampling approach for the final dataset with moderate levels of initial concern/adherence. Variance stabilizes at ~4-5 replicates (and even more tightly at ~8 replicates) so that no meaningful gains are made when sampling more replicates than this.

Fig. R4. Population-level variance in within-group standard deviations as calculated using a bootstrapping sampling approach for the final dataset with moderate levels of initial concern/adherence. Variance stabilizes at ~5 replicates so that no meaningful gains are made when sampling more replicates than this.